# Learning Semantic Anchors for Continual Generalized Category Discovery

## Abstract

Continual Generalized Category Discovery (C-GCD) aims to address the dual challenges of continual learning and generalized category discovery in open environments. This task requires the model to incrementally recognize new classes while resisting catastrophic forgetting of old classes. Existing C-GCD methods do not effectively balance the stability-plasticity dilemma, primarily due to ineffective semantic utilization and knowledge preservation, leading to poor recognition of new classes and catastrophic forgetting of old classes. To address these issues, we propose a novel C-GCD method that leverages textual descriptions and visual features to construct and optimize semantic anchors, and freeze image and text encoders to preserve general pre-trained knowledge. Extensive experiments on several datasets demonstrate that our method significantly outperforms existing C-GCD methods, effectively balancing the stability-plasticity dilemma to achieve enhanced new classes recognition and mitigated forgetting of old classes. Code is provided in the supplementary materials.

## 1 Introduction

Machine learning LeCun et al. (2015); Tu et al. (2024), especially the supervised learning paradigm, has achieved breakthroughs in tasks such as image classification and object detection. However, these methods rely on the assumption of a static and closed environment, where data are provided in offline batches with a fixed set of classes Sohn et al. (2020). This fundamentally conflicts with the dynamic and open environment of the real world Li et al. (2021); Zhu et al. (2024), where data arrive as a sequence of incremental tasks (e.g., new traffic sign types in autonomous driving) and unknown classes emerge (e.g., unknown drone morphologies). This poses a dual challenge to machine learning:

- **Continual Learning** (CL): The model needs to incrementally recognize new classes while resisting catastrophic forgetting of old classes Zhou et al. (2025).

- **Generalized Category Discovery** (GCD): The model needs to recognize unknown classes while maintaining its ability to recognize known classes Vaze et al. (2022).

Existing research often explores these two challenges in isolation: CL methods assume no unknown classes emerge Zhou et al. (2025); Zheng et al. (2025), while GCD methods require offline data batches Fan et al. (2025); Wang et al. (2025). This isolation violates the collaborative nature of continuous adaptation and active exploration in human learning processes. To address this, the task of Continual Generalized Category Discovery (C-GCD) is proposed Ma et al. (2024). This task requires the model to simultaneously address both challenges, pushing machine intelligence from a static closed paradigm to a dynamic open paradigm.

C-GCD is a novel learning paradigm designed for dynamic open environment Zhang et al. (2022); Wu et al. (2023), with its task setting illustrated in Figure 1. Specifically, the model is initially trained on a labeled dataset containing a fixed set of known classes. In subsequent stages, it processes an unlabeled data stream that includes both known and unknown classes (i.e., classes outside the set of classes in the labeled dataset). At each stage, the data stream incorporates classes from previous stages, and newly introduced unknown classes that are not included in the previous stages. Previously encountered classes (both known and unknown) are referred to as old classes, whereas

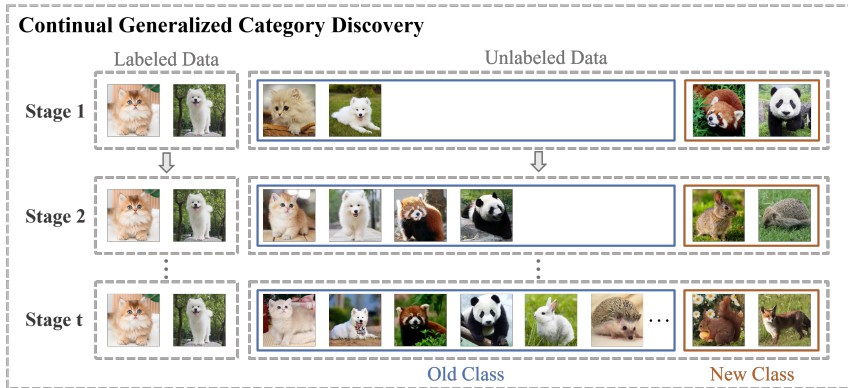

Figure 1: The environment faced by C-GCD. The model needs to incrementally recognize new classes while resisting catastrophic forgetting of old classes.

the newly introduced unknown classes are defined as new classes. The core challenge of C-GCD lies in balancing the stability-plasticity dilemma: the model needs to incrementally recognize new classes while resisting catastrophic forgetting of old classes.

However, existing C-GCD methods do not effectively balance the stability-plasticity dilemma, leading to poor recognition of new classes and catastrophic forgetting of old classes. This primarily stems from two main issues: (1) Ineffective semantic utilization. These methods represent classes as symbolic variables, which can only indicate whether a sample belongs to a certain class, thereby ignoring the rich semantic information associated with the classes Radford et al. (2021). This approach restricts the model's understanding of the deeper meanings of classes and hampers its ability to recognize new classes. (2) Ineffective knowledge preservation. These methods train multi-layer encoders in all stages, which disrupts the general feature representations learned by the pre-trained model. Furthermore, the preservation of knowledge learned in previous stages mainly relies on generating or replaying a small number of data from old classes during the training of subsequent stages Ma et al. (2024). However, these data are unable to cover the complete data distribution of the old classes, causing feature space drift during training and leading to forgetting of the old classes.

To address these issues, we propose a novel C-GCD method. This method introduces a class descriptions database and then utilizes the vision-language model to project the relevant textual descriptions of classes appearing in the current stage into the semantic space, thereby creating semantic anchors. Subsequently, we optimize these semantic anchors using visual features. Through this optimization, the model can better understand the correspondence among visual features of data and textual descriptions of classes, enhancing the model's ability to capture discriminative features of classes and thereby improving its ability to recognize new classes. Furthermore, to preserve the general feature representations learned by the pre-trained model, we freeze the image and text encoders. For knowledge preservation, we reuse the previously projected semantic anchors. Compared to replaying a few old class data, these semantic anchors provide broader coverage of the data distribution of old classes Thengane et al. (2022); Zhou et al. (2025), thereby significantly mitigating catastrophic forgetting of old classes.

Extensive experiments on standard C-GCD datasets demonstrate that our method outperforms the state-of-the-art C-GCD method, as reflected in the following aspects: (1) Enhanced recognition of new classes. Our method achieves 36.6% higher accuracy on new classes than SOTA (relative improvement) in the final incremental phase. (2) Mitigated catastrophic forgetting of old classes. Our method maintains 91.7% accuracy on old classes (vs. 75.4% for SOTA) after all incremental phases. (3) Efficient training process. Our method requires significantly less training time and GPU resources, enhancing the model's deployability in open-world settings. These results provide strong evidence that our method effectively balances the stability-plasticity dilemma in C-GCD.

To summarize, the main contributions of this paper are as follows: (1) We identify that existing C-GCD methods do not effectively balance the stability-plasticity dilemma, which primarily stems from ineffective semantic utilization and knowledge preservation, leading to poor recognition of new

classes and catastrophic forgetting of old classes. (2) We propose a novel C-GCD method, which involves utilizing textual descriptions and visual features to construct and optimize semantic anchors, and freezing the image and text encoders to maintain the general knowledge learned from the pre-trained model. (3) We conduct extensive experiments on several datasets, which demonstrate that our method significantly outperforms existing methods, effectively balancing the stability-plasticity dilemma to achieve enhanced new classes recognition and mitigated catastrophic forgetting.

## 2 RELATED WORK

**Open-World Machine Learning.** Most machine learning methods operate under the static and closed environment assumption, which frequently proves inadequate in real-world scenarios Zhu et al. (2024). To address this, methods such as continual learning (CL) Zhou et al. (2025), open-set recognition (OSR) Scheirer et al. (2013), and generalized category discovery (GCD) Vaze et al. (2022) have been developed for open-world adaptation. Although these methods make progress, they do not fully capture the complexity of open-world scenarios. To better align with real-world complexities, C-GCD is proposed to require the model to incrementally recognize continuously emerging unknown classes from unlabeled data, pushing machine intelligence from a static closed paradigm to a dynamic open paradigm. These works focus on image classification tasks, as they are crucial scenarios for testing the model's generalization ability and its capability to handle unknown classes. Therefore, we validate our proposed method on image classification tasks in this paper.

**Continual Generalized Category Discovery.** C-GCD is a novel learning paradigm designed for dynamic open environment, aiming to enable models to incrementally recognize new classes while resisting catastrophic forgetting of old classes. Although some research has explored the C-GCD task Kim et al. (2023); Zhao & Aodha (2023), they exhibit certain limitations that do not fully reflect real-world scenarios, such as considering only a limited number of incremental stages and unknown classes Wu et al. (2023), or assuming prior ratios of known samples Zhang et al. (2022). This paper adopts the environmental setting from Happy Ma et al. (2024) because it does not have the aforementioned limitations, thus offering a closer resemblance to real-world applications and a superior simulation of C-GCD task complexity. However, existing C-GCD methods do not effectively balance the stability-plasticity dilemma, which primarily stems from ineffective semantic utilization and knowledge preservation, leading to poor recognition of unknown classes and catastrophic forgetting of old classes. Therefore, we propose a novel C-GCD method to address these issues.

**Vision-Language Models.** Visual-Language Models (VLMs) have advanced significantly in recent years. These models are designed to process and understand visual and textual information, to capture the correlation between visual features and textual descriptions through extensive training on image-text pairs Li et al. (2022). They have demonstrated outstanding performance and strong generalization ability in various visual-language tasks Liu et al. (2025); Tan et al. (2025). This powerful generalization ability, coupled with their inherent ability to handle multi-modal information, makes models like CLIP suitable for open-world machine learning Radford et al. (2021). These motivate us to explore the potential of VLMs for C-GCD.

## 3 PROPOSED METHOD

In this section, we first describe the C-GCD setting. We then introduce each component of proposed C-GCD method named SAC-GCD, which includes semantic anchor construction, optimization, and expansion. The overall framework is depicted in Figure 2.

### 3.1 PROBLEM SETTING

Given a labeled dataset $D_l = \{(x_i, y_i)\}_{i=1}^{N_l}$, unlabeled datasets will arrive at different stages $t$, where $1 \leq t \leq t_{final}$, denoted as $D_u^t = \{x_i^t\}_{i=1}^{N_u^t}$. Here, $x \in \mathbb{R}^D$, where $D$ represents the feature dimension of data, and $y \in C_l$. $C_l$ represents the set of classes in $D_l$, and $C_u^t$ represents the set of classes in $D_u^t$. $C_l \subseteq C_u^t$, $C = \{C_u^t\}_{t=1}^T$ represents the set of all classes. In stage 1, $C_{\text{old}}^1 = C_l$ represents the old classes in the first stage, and $C_{\text{new}}^1 = C_u^1 \setminus C_l$ represents the new classes in the first stage. In stage $t(t \geq 2)$, $C_{\text{old}}^t = C_u^{t-1}$ represents the old classes in the $t$-th stage, and

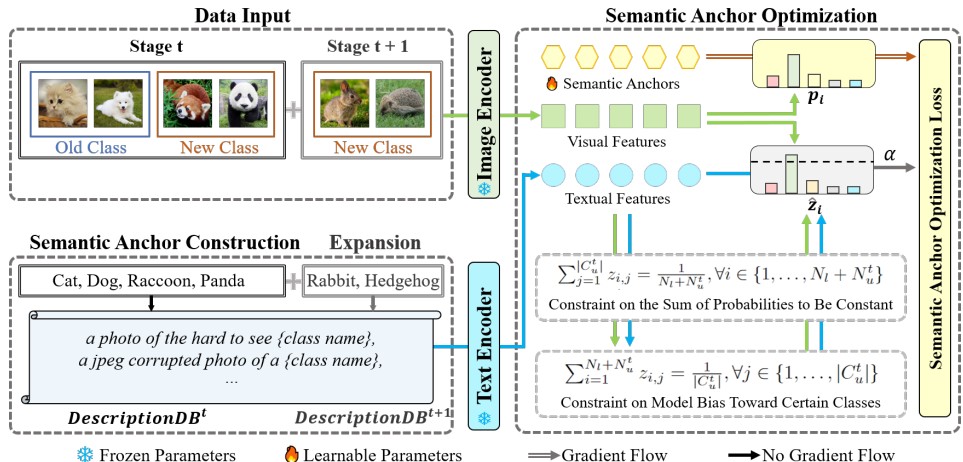

Figure 2: The proposed method framework. Our method includes semantic anchor construction, optimization, and expansion, effectively enhancing new classes recognition and mitigating catastrophic forgetting of old classes.

$C_{\text{new}}^t = C_u^t \setminus C_u^{t-1}$ represents the new classes in the $t$-th stage. $C_{\text{known}} = C_l$ represents the set of known classes, and $C_{\text{unknown}}^t = C_u^t \setminus C_l$ represents the set of unknown classes in the $t$-th stage. At stage $t$, the model is trained on $D_l \cup D_u^t$, and be evaluated on the test set $D_{test}^t = \{\bar{x}_i^t\}_{i=1}^{N_{test}^t}$. $C_{test}^t$ represents the set of classes in $D_{test}^t$, and $C_u^t = C_{test}^t$.

In this paper, we utilize deep neural networks to build our classification model $f(x; \theta_f)$, which consists of an image encoder $I(x; \theta_I) : \mathbb{R}^D \to \mathbb{R}^d$, and a text encoder $T(r; \theta_T) : \mathbb{R}^D \to \mathbb{R}^d$. Here, $D$ denotes the feature dimension of $x$, $d$ represents the feature dimension following the processing by the image encoder, and $r$ represents the textual descriptions.

## 3.2 SEMANTIC ANCHOR CONSTRUCTION

VLMs are able to capture the correlation between visual features and semantic information contained in textual descriptions associated with the classes, which makes models like CLIP particularly well-suited for C-GCD task Radford et al. (2021). Therefore, we first construct a class descriptions database and project relevant textual descriptions of classes appearing in the current stage into semantic space by VLMs to create semantic anchors. These anchors serve as initial cognition for classes and are optimized by integrating their visual features. The steps of semantic anchor construction are as follows:

**Class Description Database Construction.** There are various approaches for constructing a class description database. This paper adopts the most commonly used strategy in open-world machine learning Fan et al. (2025), which involves collecting a large number of class names from the internet and generating relevant textual descriptions using templated statements (e.g., "a photo of a [class name]"). This approach enables the construction of a diverse class description database, symbolized as $DescriptionsDB = \{r_i\}_{i=1}^q$.

**Semantic Anchor Initialization.** For stage $t$, we compute the similarity between data of $D_l \cup D_u^t$ and each class descriptions in $DescriptionsDB$. Specifically, we use the formula $\langle I(x; \theta_I), T(r; \theta_T) \rangle$ to calculate the similarity between images and textual descriptions, where $\langle \cdot, \cdot \rangle$ is dot product. For each $x_i \in D_l \cup D_u^t$, we find the most similar descriptions, count their frequencies, select the top $|C_u^t|$ most frequent ones to form $DescriptionsDB^t = \{r_i\}_{i=1}^{|C_u^t|}$ and map these descriptions into semantic space using the text encoder $T(\cdot; \theta_T)$. These serve as semantic anchors for the classes contained in $D_l \cup D_u^t$. We adopt the approach proposed by GCD Vaze et al. (2022) to estimate the number of classes $|C_u^t|$ contained in the unlabeled dataset $D_u^t$.

## 3.3 SEMANTIC ANCHOR OPTIMIZATION

However, these semantic anchors, which are textual descriptions generated using templated statements, often introduce background noise irrelevant to the classification task Lafon et al. (2024). This may potentially diminish the model's effectiveness in visual-language alignment. Therefore, we introduce a semantic anchor optimization strategy that optimizes semantic anchors by integrating their visual features, thereby enhancing the model's ability to capture discriminative features of different classes. The steps of semantic anchor optimization are as follows:

**Semantic Database Construction.** We construct a set of learnable semantic anchors for each class, denoted as $SemanticDB^t = \{w_i^t\}_{i=1}^{|C_u^t|}, w_i^t \in \mathbb{R}^d$.

**Semantic Database Optimization.** We introduce a semantic anchor optimization strategy that optimizes semantic anchors by integrating their visual features, which is conceptually similar to prompt learning Miyai et al. (2023); Lafon et al. (2024). To simplify the formula representation, in stage $t$, the data in $D_l \cup D_u^t$ is collectively referred to as $D^t = \{x_i\}_{i=1}^{N_l+N_u^t}$. The loss function of semantic anchor optimization for the purpose of optimizing $SemanticDB^t$ is as follows:

$$L = \sum_{i=1}^{N_l+N_u^t} D_{KL}(Z_i \| P_i) = \sum_{i=1}^{N_l+N_u^t} \sum_{j=1}^{C_u^t} z_{i,j} \log\left(\frac{z_{i,j}}{p_{i,j}}\right), \tag{1}$$

where $z_{i,j}$ and $p_{i,j}$ are defined as:

$$z_{i,j} = \frac{\exp\left(\langle I(x_i;\theta_I), T(r_j;\theta_T)\rangle / \tau_T\right)}{\sum_{k=1}^{|C_u^t|} \exp\left(\langle I(x_i;\theta_I), T(r_k;\theta_T)\rangle / \tau_T\right)}, \tag{2}$$

$$p_{i,j} = \frac{\exp\left(\langle I(x_i;\theta_I), w_j\rangle / \tau_I\right)}{\sum_{k=1}^{|C_u^t|} \exp\left(\langle I(x_i;\theta_I), w_k\rangle / \tau_I\right)}, \tag{3}$$

with $\tau_T$ and $\tau_I$ being temperature hyper-parameters, $\langle \cdot, \cdot \rangle$ being dot product.

**Predict Distribution Optimization.** However, the aforementioned optimization process overlooks the model bias towards different classes during training, which makes it difficult for the semantic anchors to fully exploit the semantic information of new classes. In this case, prediction bias could occur, where some new classes are incorrectly predicted as old ones Ma et al. (2024), which requires us to constrain the model so that it gives necessary attention and predictive probabilities to new classes Qian et al. (2023); Stojnic et al. (2024). The associated loss function is as follows:

$$\hat{Z} = \underset{Z}{\arg\max}\langle Z, S\rangle + \tau_T H(Z)$$

$$\text{s.t.} \begin{cases} \sum_{j=1}^{|C_u^t|} z_{i,j} = \frac{1}{N_l+N_u^t}, \forall i \in \{1, \ldots, N_l+N_u^t\}, \\ \sum_{i=1}^{N_l+N_u^t} z_{i,j} = \frac{1}{|C_u^t|}, \forall j \in \{1, \ldots, |C_u^t|\}, \\ z_{i,j} \geq 0, \forall i \in \{1, \ldots, N_l+N_u^t\}, \forall j \in \{1, \ldots, |C_u^t|\}. \end{cases} \tag{4}$$

where $Z = [z_{i,j}]_{i=1,\cdots,N_l+N_u^t}^{j=1,\cdots,|C_u^t|}$, $S = [s_{i,j}]_{i=1,\cdots,N_l+N_u^t}^{j=1,\cdots,|C_u^t|}$, and $s_{i,j} = \langle I(x_i;\theta_I), T(r_j;\theta_T)\rangle$, and $H(Z)$ computes the entropy of $Z$. The first constraint is probability normalization, which ensures that the sum of the probability distribution for each data equals a constant. The second constraint is class correction, which is commonly used in CL and GCD to prevent the model from developing a bias towards certain classes Cao et al. (2022); Wen et al. (2023). These encourage the optimized semantic anchors to better utilize the semantic information of new classes, thereby improving the model's ability to capture discriminative features.

Moreover, following semi-supervised learning principles Sohn et al. (2020); Zhang et al. (2021), the model converts high-confidence data into one-hot pseudo-labels to mitigate the impact of irrelevant classes. Therefore, we normalize $\hat{Z}$ by computing $\hat{Z} = (N_l + N_u^t) \cdot \hat{Z}$ and subsequently optimize the prediction distribution as:

$$\tilde{Z}_i = \begin{cases} e_{j^*}, & j^* = \underset{j \in \{1,\cdots,|C_u^t|\}}{\arg\max} \{\hat{z}_{i,j}\}, \hat{z}_{i,j^*} > \alpha, \\ \hat{Z}_i, & \text{otherwise.} \end{cases} \tag{5}$$

where $e_{j^*} \in \{0,1\}^{|C_u^t|}$ is a one-hot vector with the $j^*$-th element being 1 and $\alpha$ is a predefined threshold. $\tilde{Z}_i$ will replace $Z_i$ in Eqn. (1), thereby updating the loss function for semantic anchor optimization.

### 3.4 SEMANTIC ANCHOR EXPANSION

In the next stage $t + 1$, new classes will appear in $D_u^{t+1}$. Therefore, we need to generate semantic anchors for these new classes to expand the semantic anchor database. Moreover, due to the addition of new classes, the data distribution in stage $t+1$ will change compared to stage $t$ Zheng et al. (2025). Consequently, we need to further optimize the semantic anchors in $SemanticDB^t$. Specifically, in stage $t + 1$, we construct $DescriptionsDB^{t+1}$ through semantic anchor initialization, and further execute the steps of semantic database construction, semantic database optimization, and prediction distribution optimization, thereby generating the corresponding $SemanticDB^{t+1}$ for stage $t + 1$. Compared to replaying a few old class data, these semantic anchors provide broader coverage of the data distribution of old classes Thengane et al. (2022), thereby significantly mitigating catastrophic forgetting of old classes.

### 3.5 INFERENCE

In stage $t$, we train the model on $D_l \cup D_u^t$ to acquire $SemanticDB^t$ and test on $D_{test}^t$. We follow Stojnic et al. (2024) to concatenate the visual features $I(\bar{x}_i^t; \theta_I)$ of $D_{test}^t$ with the semantic anchors $SemanticDB^t$, then execute the bimodal label propagation algorithm, thereby obtaining the predicted label $Result_i$ for each $\bar{x}_i^t \in D_{test}^t$.

## 4 EXPERIMENTS

In this section, we conduct a comprehensive evaluation of our method. The experimental results and detailed analyses demonstrate the superiority of our method over existing C-GCD methods.

### 4.1 EXPERIMENTAL SETUP

**Datasets.** We conduct experiments on four standard C-GCD datasets: CIFAR100 (C100) Krizhevsky & Hinton (2009), ImageNet-100 (IN-100) Deng et al. (2009), Tiny-ImageNet (Tiny) Chrabaszcz et al. (2017) and CUB Wah et al. (2011), which contain 100, 100, 200 and 200 classes, respectively. To ensure the fairness of the experiment, we conduct our experiments using the data partitioning

Table 1: Dataset splits of C-GCD setting. img-class represents the number of images in each class, img-new represents the number of images in each new class, and img-old represents the number of images in each old class.

| Datasets | Labeled | | Unlabeled Each Stage $t$ ($t$=1,...,5) | | |
|---|---|---|---|---|---|
| | $|C_l|$ | img-class | $|C_u^t \setminus C_l|$ | img-new | img-old |
| C100 | 50 | 400 | 10 | 400 | 25 |
| IN100 | 50 | ~1000 | 10 | 1000 | 60 |
| Tiny | 100 | 400 | 20 | 400 | 25 |
| CUB | 100 | ~25 | 20 | 25 | 5 |

method described in Happy Ma et al. (2024). We divide the classes into known and unknown classes using the same approach, with 50% of the classes regarded as known and the remaining 50% as unknown. We choose a portion of data from the known classes to construct the dataset $D_l$. During stage $t = 1$, known classes are termed as old classes, new unknown classes are termed as new classes. We select a portion of data from both old and new classes to build the dataset $D_u^t$. During stage $t(t \geq 2)$, we select a portion of data from both old classes (i.e., all classes that appeared in stage $t - 1$) and new classes (i.e., classes that newly appear in stage $t$) to build the dataset $D_u^t$. $D_{test}^t$ includes all classes that appear in stage $t$, including both old and new classes. This procedure is applied consistently to all compared methods. Detailed dataset information is provided in Table 1.

**Compared Methods.** We compare our method with several methods including K-means Ikotun et al. (2023), VanillaGCD Vaze et al. (2022), SimGCD Wen et al. (2023), SimGCD+LwF Li & Hoiem (2018), FRoST Roy et al. (2022), GM Zhang et al. (2022), MetaGCD Wu et al. (2023), and Happy Ma et al. (2024). These selected methods collectively represent the latest advancements

Table 2: Classification accuracy (%) of compared methods on All, Old and New classes. Bold font indicates the best classification accuracy achieved on the corresponding dataset.

| Datasets | Methods | Stage-1 | | | Stage-2 | | | Stage-3 | | | Stage-4 | | | Stage-5 | | |
|---|---|---|---|---|---|---|---|---|---|---|---|---|---|---|---|---|
| | | All | Old | New | All | Old | New | All | Old | New | All | Old | New | All | Old | New |
| C100 | KMeans | 40.27 | 41.76 | 32.80 | 37.14 | 38.33 | 30.00 | 36.20 | 37.63 | 26.20 | 36.66 | 38.30 | 23.50 | 35.69 | 36.79 | 25.80 |
| | VanillaGCD | 72.32 | 78.50 | 41.40 | 67.04 | 72.50 | 34.30 | 57.99 | 62.26 | 28.10 | 56.60 | 59.55 | 33.00 | 51.36 | 53.70 | 30.30 |
| | SimGCD | 73.37 | 86.44 | 8.00 | 62.56 | 72.43 | 3.30 | 54.17 | 61.61 | 2.10 | 47.62 | 53.37 | 1.60 | 43.53 | 47.86 | 4.60 |
| | SimGCD+ | 75.93 | **87.04** | 20.40 | 67.07 | 75.33 | 17.50 | 58.45 | 64.33 | 17.30 | 54.31 | 58.71 | 19.10 | 50.49 | 53.90 | 19.80 |
| | FRoST | 76.87 | 79.58 | 63.30 | 65.31 | 68.88 | 43.90 | 58.01 | 61.09 | 36.50 | 49.27 | 50.90 | 36.20 | 48.03 | 48.17 | 46.80 |
| | GM | 76.58 | 79.80 | 60.50 | 71.10 | 74.52 | 50.60 | 63.51 | 68.16 | 31.00 | 59.74 | 62.51 | 37.60 | 54.11 | 54.74 | 48.40 |
| | MetaGCD | 76.12 | 83.60 | 38.70 | 69.40 | 72.82 | 48.90 | 61.95 | 65.76 | 35.30 | 58.22 | 61.21 | 34.30 | 55.78 | 58.47 | 31.60 |
| | Happy | **80.40** | 85.26 | 56.10 | **74.13** | **78.27** | 49.30 | 68.23 | **70.86** | 49.80 | 62.26 | 63.75 | 50.30 | 59.99 | 60.96 | 51.30 |
| | SAC-GCD | 75.07 | 74.78 | **76.50** | 70.64 | 72.48 | **59.60** | **68.44** | 69.61 | **60.20** | **67.77** | **67.44** | **70.40** | **68.87** | **69.72** | **61.20** |
| IN100 | KMeans | 54.90 | 57.04 | 44.20 | 54.73 | 56.37 | 44.90 | 54.67 | 56.66 | 40.80 | 54.63 | 56.25 | 41.70 | 53.92 | 56.18 | 33.60 |
| | VanillaGCD | 70.13 | 72.92 | 56.20 | 69.37 | 73.47 | 44.80 | 68.50 | 70.63 | 53.60 | 65.56 | 67.85 | 47.20 | 64.54 | 67.44 | 38.40 |
| | SimGCD | 79.67 | 91.68 | 19.60 | 70.23 | 78.83 | 18.60 | 61.90 | 67.43 | 23.20 | 56.67 | 60.92 | 22.60 | 52.90 | 56.40 | 21.40 |
| | SimGCD+ | 83.07 | 95.16 | 22.60 | 74.57 | 83.47 | 21.20 | 67.60 | 73.57 | 25.80 | 62.09 | 66.83 | 24.20 | 57.62 | 61.47 | 23.00 |
| | FRoST | 87.50 | 92.96 | 60.20 | 79.63 | 83.37 | 57.20 | 76.78 | 77.00 | 75.20 | 66.18 | 68.65 | 46.40 | 63.82 | 66.40 | 40.60 |
| | GM | 89.53 | 95.04 | 62.00 | 82.34 | 86.93 | 54.80 | 77.97 | 79.17 | 69.60 | 72.80 | 74.65 | 58.00 | 71.08 | 71.76 | 65.00 |
| | MetaGCD | 75.27 | 78.20 | 60.60 | 73.79 | 75.93 | 54.90 | 69.35 | 72.20 | 49.40 | 67.22 | 70.10 | 44.20 | 66.68 | 69.31 | 43.00 |
| | Happy | 91.20 | **95.36** | 70.40 | 87.83 | 90.83 | 69.80 | 85.22 | 86.40 | 77.00 | 81.93 | 83.00 | 73.40 | 78.58 | 79.11 | 73.80 |
| | SAC-GCD | **92.50** | 92.00 | **95.00** | **91.26** | **91.40** | **90.40** | **89.78** | **90.20** | **86.80** | **88.00** | **88.40** | **84.80** | **88.62** | **88.67** | **88.20** |
| Tiny | KMeans | 35.42 | 35.46 | 35.20 | 34.99 | 35.75 | 30.40 | 34.80 | 36.07 | 25.90 | 34.77 | 35.90 | 24.90 | 34.62 | 35.63 | 25.50 |
| | VanillaGCD | 55.93 | 58.92 | 41.00 | 54.96 | 58.58 | 33.20 | 52.82 | 55.74 | 32.40 | 48.81 | 51.46 | 27.60 | 45.94 | 48.06 | 26.90 |
| | SimGCD | 66.95 | 79.94 | 2.00 | 57.81 | 66.98 | 2.80 | 52.70 | 59.83 | 2.77 | 45.01 | 50.29 | 2.80 | 41.59 | 45.79 | 3.80 |
| | SimGCD+ | 70.38 | 81.80 | 13.30 | 62.47 | 70.75 | 12.80 | 54.55 | 60.46 | 13.20 | 47.98 | 52.49 | 11.90 | 42.98 | 46.46 | 12.70 |
| | FRoST | 75.15 | 78.56 | 58.10 | 65.64 | 67.83 | 52.50 | 51.32 | 54.31 | 30.40 | 48.22 | 52.14 | 16.90 | 40.15 | 42.73 | 16.90 |
| | GM | 76.42 | **82.40** | 46.50 | 68.87 | 73.82 | 39.20 | 58.68 | 63.43 | 25.40 | 52.86 | 57.21 | 18.10 | 46.90 | 50.62 | 13.40 |
| | MetaGCD | 60.88 | 64.90 | 40.80 | 57.20 | 61.03 | 34.02 | 54.36 | 57.19 | 34.60 | 50.83 | 53.59 | 28.80 | 48.14 | 50.16 | 30.00 |
| | Happy | **78.85** | **82.40** | 61.10 | **71.34** | **76.18** | 42.30 | 64.68 | 68.70 | 36.50 | 58.49 | 60.64 | 41.30 | 54.56 | 56.66 | 35.70 |
| | SAC-GCD | 70.18 | 68.56 | **78.30** | 66.49 | 67.05 | **63.10** | **65.55** | 64.94 | **69.80** | **64.08** | **63.35** | **69.90** | **63.50** | **62.77** | **70.10** |
| CUB | KMeans | 32.54 | 30.76 | 41.18 | 31.19 | 30.53 | 35.20 | 29.28 | 27.46 | 42.09 | 29.19 | 28.13 | 37.61 | 28.17 | 27.01 | 38.53 |
| | VanillaGCD | 64.47 | 67.06 | 51.93 | 58.15 | 60.65 | 42.91 | 54.10 | 56.40 | 37.91 | 49.98 | 51.33 | 39.32 | 46.84 | 46.58 | 49.14 |
| | SimGCD | 73.84 | 84.54 | 22.02 | 63.36 | 72.35 | 8.58 | 55.63 | 61.95 | 11.13 | 49.31 | 54.55 | 7.86 | 44.72 | 48.69 | 9.25 |
| | SimGCD+ | 75.62 | **85.55** | 25.97 | 65.32 | 73.93 | 13.68 | 57.40 | 63.28 | 16.26 | 51.11 | 55.72 | 14.72 | 45.79 | 49.29 | 14.28 |
| | FRoST | 77.03 | 83.95 | 43.53 | 50.77 | 53.46 | 34.33 | 46.42 | 49.31 | 26.09 | 39.40 | 41.47 | 23.08 | 34.55 | 35.12 | 29.45 |
| | GM | 76.17 | 80.23 | 56.51 | 67.91 | 73.38 | 34.58 | 61.12 | 66.53 | 23.00 | 51.96 | 54.40 | 30.10 | | | |
| | MetaGCD | 67.08 | 70.21 | 51.92 | 60.77 | 62.39 | 50.86 | 57.53 | 59.33 | 37.78 | 51.90 | 52.22 | 49.40 | 49.60 | 49.96 | 46.38 |
| | Happy | **81.40** | 85.06 | 63.70 | **74.27** | **76.03** | 63.57 | 67.09 | **71.06** | 39.13 | 62.25 | 63.83 | 49.74 | 59.39 | 60.49 | 49.52 |
| | SAC-GCD | 72.71 | 74.34 | **64.35** | 70.53 | 70.30 | **71.86** | **68.62** | 69.25 | **64.11** | **65.87** | **67.47** | **53.52** | **64.46** | **64.08** | **67.79** |

in the field of C-GCD. For detailed implementation of these methods, please refer to the relevant literature such as Happy.

**Evaluation Protocol.** Following Happy Ma et al. (2024), we use the clustering accuracy (ACC) to evaluate the performance of our method at each stage. Specifically, $ACC$ is calculated on $D_{test}^t$ as illustrated in the following equation:

$$ACC = \frac{1}{N_{test}^t} \sum_{i=1}^{N_{test}^t} \mathbb{I}(\text{label}_i = OP\left(\text{Result}_i\right)). \tag{6}$$

In the equation, $\text{label}_i$ represents the actual class label for $\bar{x}_i^t \in D_{test}^t$, which is only provided during the testing phase. $OP$ stands for the optimal permutation that aligns $\text{Result}_i$ with $\text{label}_i$. Our evaluation includes calculating $ACC$ for all data (All), data from old classes (Old) in $D_{test}^t$, and data from new classes (New) in $D_{test}^t$.

**Implementation Details.** We utilize the ViT-B/16 model Dosovitskiy et al. (2021), pre-trained with CLIP Radford et al. (2021), as the backbone network for both image and text encoders, and freeze its parameters. Concurrently, we leverage the templated statements proposed by CLIP to construct the class description database. Furthermore, all compared methods assume that the number of classes in each stage is known, thus we also follow this assumption in our method. To ensure the generalization of our method, unless otherwise specified, we adopt the following unified hyperparameter settings across all datasets: $\tau_T = 0.01$, $\tau_I = 0.04$, $\alpha = 0.5$. All experiments are conducted on a single NVIDIA A6000 GPU. For more implementation details, please refer to the code.

## 4.2 Main Results

The classification accuracy of different methods across various datasets is provided in Table 2. We report the average maximum classification accuracy from three runs for our method, whereas the remaining results are obtained from Happy Ma et al. (2024). The experimental results demonstrate that our method outperforms the state-of-the-art C-GCD method Happy, effectively balancing the stability-plasticity dilemma inherent in C-GCD. This validates the superiority of our method, which is reflected in the following three aspects:

**Enhanced New Classes Recognition.** A crucial task for C-GCD is to incrementally recognize new classes. We evaluate the model's performance on recognizing new classes across all stages. Experimental results show that our method significantly outperforms the Happy in all stages, especially in the final stage, where the average improvement in recognizing new classes across different datasets reaches 36.6% (relative improvement). This demonstrates that our method has a significant advantage in recognizing new classes.

**Mitigated Catastrophic Forgetting.** Another crucial task for C-GCD is to resist catastrophic forgetting of old classes. To verify this, we evaluate the model's ability to recognize all classes in the final stage, and compare it to its performance in the initial stage. The results show that the Happy retained only 75.4% of its initial performance, while our method achieved 91.7%. Moreover, starting from the stage 3, our method consistently outperformed Happy, and in the final stage, the model's ability to recognize all classes improved by an average of 13.0%. This demonstrates that our method better retains knowledge of old classes during the incremental learning process, effectively mitigating catastrophic forgetting.

**Efficient Training Process.** A key metric for applying C-GCD to practical tasks is efficiency and low resource consumption. Our method achieves this by training only the semantic anchors, thereby enabling efficiency and low resource usage. Specifically, our method completes training across all stages in just half a minute, whereas Happy requires several hours. This significant improvement in efficiency makes our method more suitable for open environments.

## 4.3 Analyses and Discussions

In this section, we will conduct detailed analyses and discussions around aforementioned three aspects, demonstrating the effectiveness of the strategies employed in our method and their superiority in the C-GCD task. Additional analysis experiments will be provided in the appendix.

**Analysis of effectiveness of different methods for C-GCD tasks.** The key tasks for C-GCD are enhancing new classes recognition and mitigating catastrophic forgetting. In order to analyze the effectiveness of different methods on these tasks, GM Zhang et al. (2022) designs two metrics $M_d$ and $M_f$, which are defined as follows:

$$M_d = ACC_{New}^{t_{final}}. \tag{7}$$

$$M_f = \max_t \left\{ ACC_{Old}^0 - ACC_{Old}^t \right\}, \tag{8}$$

We use $M_d$ and $M_f$ to evaluate different C-GCD methods and the relevant experimental results are shown in Table 3. The effectiveness of enhancing new classes recognition is reflected by higher $M_d$, while the effectiveness of mitigating catastrophic forgetting is reflected by lower $M_f$. The experimental results demonstrate that our method significantly outperforms the state-of-the-art methods.

Table 3: Performance comparison between C-GCD methods on $M_d$ and $M_f$.

| Datasets | Methods | $M_d \uparrow$ | $M_f \downarrow$ |
|---|---|---|---|
| | GM | 65.00 | 23.28 |
| IN100 | Happy | 73.80 | 16.25 |
| | SAC-GCD | **88.20** | **3.33** |
| | GM | 13.40 | 31.78 |
| Tiny | Happy | 35.70 | 25.74 |
| | SAC-GCD | **70.10** | **5.79** |

Table 4: Comparison of the training times and GPU resources of different methods.

| Datasets | Methods | Training time | GPU resources |
|---|---|---|---|
| C100 | Happy | 22068.83 s | 9859.37 MB |
| | SAC-GCD | 12.26 s | 1076.65 MB |
| CUB | Happy | 8468.84 s | 9862.36 MB |
| | SAC-GCD | 9.74 s | 231.08 MB |

**Analysis of efficiency of different C-GCD methods.** A key metric for applying C-GCD methods to practical tasks is efficiency and low resource consumption. Our method achieves this by freezing

the parameters of the image and text encoders, requiring only a single pass through the image-text encoder for image and textual descriptions, and then optimizing only the semantic anchors. This method enables high efficiency and low resource consumption. We compare the training time and GPU resources required by the Happy and our method on different datasets. As shown in Table 4, our method requires significantly less training time and GPU resources than Happy. Due to the excessive runtime of Happy on the other two datasets, the results are not recorded. This significant improvement in efficiency makes our method more suitable for resource-constrained open environments that require continuous learning.

**Analysis of the sensitivity of hyperparameters.** To ensure the good generalization ability of our method, we adopt a unified set of hyperparameters across all datasets: $\tau_T = 0.01$, $\tau_I = 0.04$, and $\alpha = 0.5$. To verify the robustness of our method to different values of these hyperparameters, we adjust $\tau_T$, $\tau_I$, and $\alpha$ individually while keeping all other parameters constant. We demonstrate the performance of the model in the final stage, and the relevant experimental results are shown in Figure 3. These analyses demonstrate that our method is robust to different values of hyperparameters $\tau_T$, $\tau_I$, and $\alpha$.

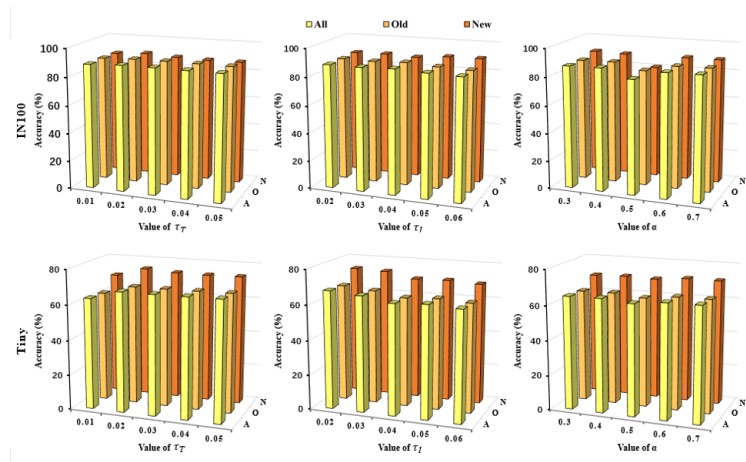

Figure 3: Performance of SAC-GCD with different values of hyperparameters.

## 5 CONCLUSION

In this paper, we propose a novel method for C-GCD. This method comprises the construction, optimization, and expansion of semantic anchors, effectively balancing the stability-plasticity dilemma, thereby effectively enhancing the recognition of new classes and mitigating catastrophic forgetting of old classes. Experimental results across multiple datasets show that our method significantly outperforms the state-of-the-art C-GCD method across multiple evaluation metrics. This not only fully demonstrates the effectiveness of our method but also clearly reveals the substantial potential of visual-language models for C-GCD tasks. We are convinced that this method can serve as a foundational framework for future research. Moreover, this paper primarily discusses classification tasks, while future work can extend the C-GCD learning paradigm to areas such as object detection and segmentation.

## 6 REPRODUCIBILITY STATEMENT

This paper introduces a novel C-GCD method. The core code of the algorithm is included in the supplementary materials and will be made publicly available on GitHub upon acceptance of the paper. Additionally, both the datasets and pre-trained model weights used in this paper are publicly accessible, enabling researchers to easily download them from their respective platforms and reproduce our method.

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

# A  APPENDIX

**The Use of Large Language Models**

In this paper, we utilize Large Language Models (LLMs) as an auxiliary tool, solely for grammar correction and language polishing. LLMs are not involved in research ideation, experimental design, data analysis, or the writing of core content in the paper.

**Analysis of effectiveness of semantic anchor optimization.** We adopt the templated statements (e.g., "a photo of a [class name]") to generate textual descriptions relevant to each class, and utilize the text encoder to generate corresponding semantic anchors. Subsequently, we optimize these semantic anchors using visual features, thereby enhancing the model's ability to capture discriminative features of different classes. To validate the effectiveness of the semantic anchor optimization strategy, we compare the model's performance on the C-GCD task under two conditions: one without semantic anchor optimization and one with it. Figure 4 shows the relevant experimental results, confirming that semantic anchor optimization significantly improves the model's performance on all evaluation metrics, thereby validating the effectiveness of the strategy.

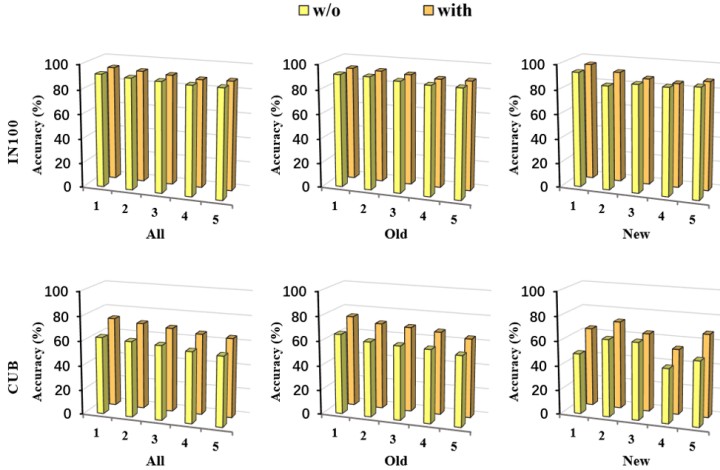

Figure 4: Performance comparison between methods without and with semantic anchor optimization.

**Analysis of effectiveness more realistic C-GCD scenarios.** In the aforementioned experiments, we use a more challenging C-GCD task setting, which is similar to that of Happy. Specifically, this setting includes: more continuing phases and new classes to be recognized, without using a replay mechanism, without storing previous data, an unknown proportion of data from new classes, and a significantly smaller number of data from old classes compared to those from new ones in each phase. However, Happy still operates under the assumption of using separate training and test sets. This is largely due to the long training times of Happy, which require isolating the training process from the test data. Thanks to the efficiency of our method, we can employ transductive learning, where the model can access test data during training and utilize this information to optimize semantic anchors, with the final evaluation being performed on the test set. This setting is similar in spirit to test-time adaptation Gao et al. (2024), which also leverages test data information to adjust model parameters. Figure 5 shows the relevant experimental results, where SAC-GCD+ represents the experiments conducted under more realistic C-GCD scenarios. Experiments show that our method achieves improved performance in this setting.

**Analysis of effectiveness of enhancing new classes recognition.** A key task for C-GCD is to effectively recognize new classes. To verify the effectiveness of our method on this task, we evaluate the model's ability to recognize new classes at different stages. Figure 6 shows the relevant experimental results. Compared to the state-of-the-art C-GCD method Happy, our method significantly outperforms Happy in all stages. In the final stage, on different datasets, our method achieves a 36.6% average improvement in recognizing unknown classes and a 13.0% average improvement in recog-

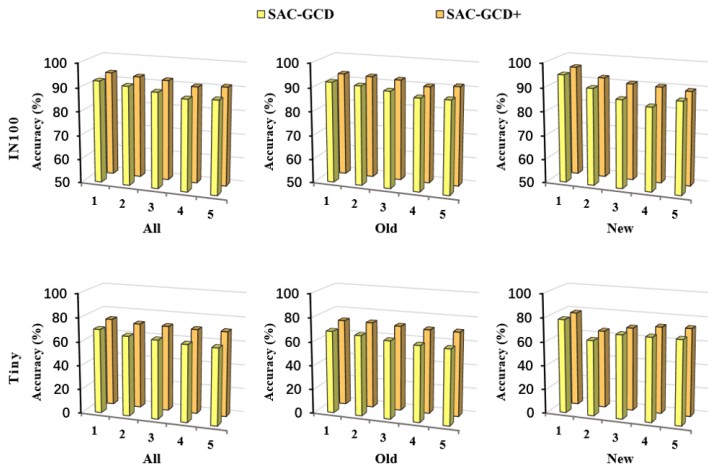

Figure 5: Performance comparison between methods in basic and more realistic C-GCD scenarios.

nizing all classes. This fully confirms that our method effectively balances the stability-plasticity dilemma between known and unknown classes.

**Analysis of effectiveness of mitigating catastrophic forgetting.** Another crucial task for C-GCD is to resist catastrophic forgetting of old classes. To verify the effectiveness of our method on this task, we evaluate the model's ability to recognize all classes at different stages and compare it with the performance of the initial stage ($ACC_{All}^t/ACC_{All}^1$). Figure 7 shows the relevant experimental results. Compared with the state-of-the-art C-GCD method Happy, our method can effectively resist catastrophic forgetting of old classes. In the final stage, on different datasets, Happy only retains an average of 75.4% of its initial performance, while our method reaches 91.7%. In addition, our method achieves a 13.0% average overall performance improvement compared to Happy. This fully confirms that our method effectively balances the stability-plasticity dilemma between old and new classes.

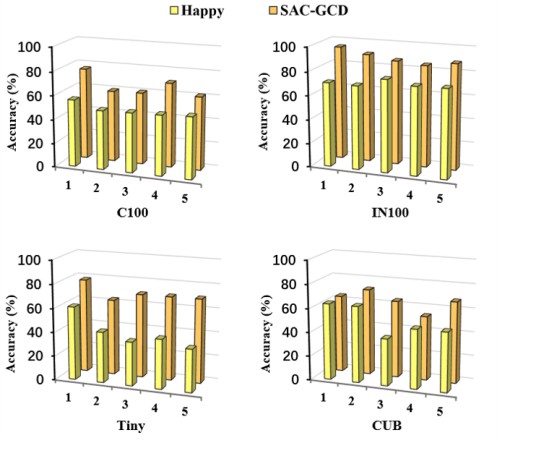
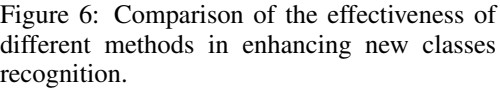
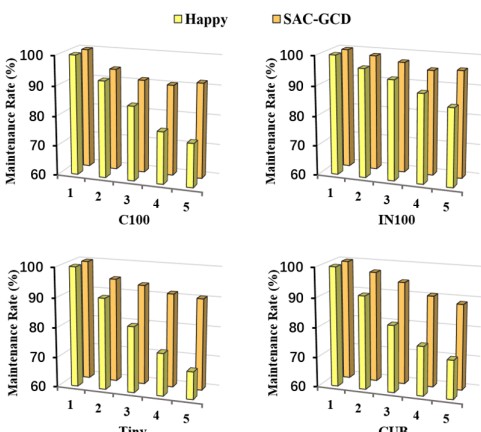

Figure 6: Comparison of the effectiveness of different methods in enhancing new classes recognition.

Figure 7: Comparison of the effectiveness of different methods in mitigating catastrophic forgetting.

