# OpenReview forum: "Learning Semantic Anchors for Continual Generalized Category Discovery"
_ICLR.cc/2026/Conference — ICLR 2026 Conference Withdrawn Submission_

### Official Review · Reviewer_5ARW · 2025-10-28

**Soundness:** 2
**Presentation:** 3
**Contribution:** 2
**Rating:** 2
**Confidence:** 4

**Summary:**

This paper tackles the challenging task of Continual Generalized Category Discovery (C-GCD), which jointly considers continual learning and open-world category discovery. The authors propose to leverage CLIP’s pre-trained image and text encoders to provide “semantic anchors,” using textual descriptions and visual features to guide incremental learning without catastrophic forgetting. Both the image and text encoders are frozen to preserve general semantic knowledge, and only the semantic anchor module is optimized across incremental stages.

**Strengths:**

1. The paper correctly identifies that most existing C-GCD methods struggle to balance the stability–plasticity trade-off and fail to effectively use pre-trained semantic knowledge.

2. The idea of constructing and optimizing semantic anchors while freezing the CLIP backbone is conceptually simple and practically feasible.

**Weaknesses:**

1. Using CLIP as the feature extractor is problematic because CLIP itself already performs zero-shot classification based on extensive pre-trained knowledge. This introduces additional external information, which contradicts the core goal of C-GCD — discovering and recognizing unknown categories from data rather than relying on prior semantics.
Moreover, it is not evident whether the proposed approach actually learns or adapts from new data in the continual setting, limiting its adaptability and novelty compared to existing works.

2. The semantic anchors are built from an arbitrarily defined textual vocabulary. This design introduces strong human bias and heavy prior knowledge, and it cannot guarantee coverage of all possible categories, which weakens the claim of “generalized” category discovery.

3. No experiment removes the Semantic Anchor Optimization component. Without such an ablation, it is impossible to verify its actual contribution. The current results could simply reflect CLIP’s inherent zero-shot classification ability rather than the proposed method’s innovation.

4. The paper does not include comparisons with several recent and relevant works, such as PromptCCD and VB-CGCD, which makes it difficult to assess the claimed superiority and novelty.

5. The provided code appears inconsistent with the method described in the paper, which is confusing and raises concerns about reproducibility and reliability.

Overall, the work shows limited innovation beyond using CLIP within the C-GCD framework. Compared to this paper [1], it lacks convincing evidence of novel contribution.

[1] Yuxin Fan, Junbiao Cui, Jiye Liang. Learning Textual Prompts for Open-World Semi-Supervised Learning. Proceedings of the IEEE/CVF Conference on Computer Vision and Pattern Recognition (CVPR), 2025, pp. 14756-14765

**Questions:**

1. Could you conduct an ablation study to show the performance change after removing the Semantic Anchor Optimization module? Specifically, how does the model perform when directly using CLIP features and the constructed anchors for classification without optimization?

2. Could you compare your method with more recent and relevant baselines, such as PromptCCD[1] and VB-CGCD[2]?

3. How does your method handle the situation where new classes emerge that are not covered by any of the constructed anchors?

[1] Cendra, F.J., Zhao, B., Han, K.. PromptCCD: Learning Gaussian Mixture Prompt Pool for Continual Category Discovery. ECCV 2024.
[2] Hao Dai, Jagmohan Chauhan. Continual Generalized Category Discovery: Learning and Forgetting from a Bayesian Perspective. Proceedings of the 42nd International Conference on Machine Learning, PMLR 267:11884-11903, 2025.

---

### Official Review · Reviewer_6Bo9 · 2025-10-31

**Soundness:** 3
**Presentation:** 3
**Contribution:** 2
**Rating:** 4
**Confidence:** 4

**Summary:**

The paper proposes SAC-GCD, a CLIP-based approach for Continual Generalized Category Discovery that builds semantic anchors from text prompts, optimizes them with visual features, freezes the image/text encoders, and performs bimodal label propagation at inference. Experiments on CIFAR-100, ImageNet-100, Tiny-ImageNet, and CUB report notable gains over prior works and large efficiency improvements.

**Strengths:**

1. The paper tackles an important problem in continual generalized category discovery (C-GCD) using a simple and modular framework that combines frozen encoders with learnable anchors.
2. The proposed method achieves strong performance across multiple datasets and stages, showing improved new-class accuracy, reduced forgetting, and significantly shorter training time.
3. The writing and figures are clear and easy to follow, and the task setting is well described and motivated.

**Weaknesses:**

- The main components, including prompt or anchor learning for VLMs and label propagation, are already well studied. The paper would benefit from a clearer positioning relative to existing prompt-learning methods, as well as recent CLIP-based label propagation approaches, rather than only citing them. It remains unclear what is fundamentally new in the optimization beyond existing prompt-tuning and distribution-balancing techniques.
- SAC-GCD leverages CLIP ViT-B/16 with frozen encoders; not all baselines do. Ensure apples-to-apples comparisons (same backbone/pretraining, frozen vs. trainable policies) and include CLIP-based strong baselines with prompt tuning. Some reported gains (e.g., very high new-class accuracy and drastic speedups) look unusually large and warrant careful controls.
- Some algorithmic details are missing. For example, how exactly is Eq. (4) optimized per batch/epoch? Provide algorithmic steps, runtime, and ablation without that constraint, the practical procedure for anchor selection and expansion, and how memory and computation scale as classes increase.

**Questions:**

- Can you report comparisons against CLIP-prompt baselines (e.g., learnable prompts / visual prompts) under identical frozen-encoder setups?
- What is the per-stage memory footprint as SemanticDB grows, and how does inference latency scale with the number of anchors?

---

### Official Review · Reviewer_MGMg · 2025-11-01

**Soundness:** 2
**Presentation:** 1
**Contribution:** 2
**Rating:** 2
**Confidence:** 5

**Summary:**

This paper proposes a vision-language-based framework named SAC-GCD. The goal is to address the stability-plasticity dilemma in continual learning scenarios where new categories emerge over time (Generalized Category Discovery).
The method constructs semantic anchors by leveraging class textual descriptions projected via a pre-trained CLIP model. These anchors are optimized using visual features to better align the visual-text space. To prevent forgetting, both the image and text encoders are frozen, and previously learned semantic anchors are reused across stages. Experiments on standard datasets (CIFAR100, IN-100, Tiny-ImageNet, CUB) reportedly outperform prior C-GCD baselines in both new-class recognition and mitigating forgetting, while improving efficiency by avoiding retraining of large encoders.

**Strengths:**

+ The paper provides a coherent description of the C-GCD problem setting and the associated challenges of balancing plasticity and stability.
+ Freezing CLIP encoders and optimizing only semantic anchors significantly reduces training time and computational cost, which is practically appealing for open-world scenarios.
+ Evaluations across multiple datasets with comparisons with several baselines demonstrate consistent performance gains in both accuracy and efficiency.

**Weaknesses:**

- The C-GCD task itself has been well established. The presented framework mostly reuses existing components, such as frozen CLIP embeddings and semantic projections, without introducing a fundamentally new formulation or learning objective.
- The core mechanism (semantic anchors optimized with visual features) is conceptually similar to existing proxy-based or anchor-based representation learning methods, for example, Proxy Anchor-based Unsupervised Learning for Continuous Generalized Category Discovery, Kim et al., ICCV 2023. The primary modification is the inclusion of CLIP-derived text features, which is a rather straightforward extension rather than a novel learning principle.
- The paper does not sufficiently disentangle the contribution of each design choice, (1) the use of CLIP text features vs. standard learned anchors, (2) the necessity of freezing encoders, (3) the specific anchor optimization loss. A quantitative ablation study is needed.
- The text flow is often repetitive, overly verbose, and lacks linguistic smoothness. Citations are sometimes incorrectly formatted (missing parentheses, inconsistent year styles), and certain sentences are grammatically awkward or redundant. This affects clarity and readability.

**Questions:**

1. How does the proposed “semantic anchor optimization” differ algorithmically and conceptually from prior proxy/anchor-based GCD methods, particularly Kim et al. (ICCV 2023) and Park et al. (ECCV 2024)? A direct comparison or visualization of the optimization process would help justify novelty.
Kim et al. Proxy Anchor-based Unsupervised Learning for Continuous Generalized Category Discovery, Park et al. Online Continuous Generalized Category Discovery
2. Can the authors include results showing performance with and without CLIP text features (or with different VLMs) to quantify the contribution of text-guided semantic information?
3. Since CLIP embeddings are frozen, how does the method adapt when domain shift occurs, for example, medical or industrial datasets where CLIP’s prior fails? Does the performance drop significantly?

---

### Official Review · Reviewer_JiZD · 2025-11-01

**Soundness:** 1
**Presentation:** 1
**Contribution:** 1
**Rating:** 0
**Confidence:** 5

**Summary:**

The paper tackles Continual Generalized Category Discovery (C-GCD) and proposes SAC-GCD, a CLIP-based pipeline. At every stage, the method (i) builds a class-description database from internet class names and CLIP-style templates, (ii) uses current-stage data to pick the top $|C_u^t|$ descriptions, (iii) turns them into semantic anchors and optimizes them by matching CLIP text similarities to learnable visual anchors with a KL loss, and (iv) finally performs bimodal label propagation over test features and anchors for prediction. The authors claim consistent gains over C-GCD baselines on CIFAR-100, ImageNet-100, Tiny-ImageNet, and CUB compared to other C-GCD methods.

**Strengths:**

1. **Clear motivation**: the paper claims that current C-GCD methods rarely exploit class-level semantics and often retrain multi-layer encoders, which can hurt old classes. SAC-GCD is a multimodal alternative.
2. **Lightweight training**: freezing CLIP and only learning stage-wise semantic anchors is attractive from a deployment perspective.

**Weaknesses:**

**(A) Fairness of comparison is not established.**
Although the paper says “we follow the partitioning in Happy … to ensure fairness,” the *actual* pipeline is much stronger than many of the reported baselines: it uses a frozen CLIP as backbone and at test time concatenates test features with semantic anchors and runs bimodal label propagation (LP). Most baselines in Table 2 were not conducted in CLIP or designed  for this inference mode, and the paper appears to reuse their reported numbers rather than rerunning them under “frozen CLIP + LP.” This makes it hard to tell whether gains come from the proposed semantic-anchor machinery or simply from (i) using CLIP, and (ii) using a transductive LP head. A minimally fair comparison would require:

* baselines rerun on the **same CLIP-ViT encoder**,
* baselines evaluated with the **same LP inference**, or
* an ablation “SAC-GCD w/o LP”.
  Without this, the headline table is not persuasive.

**(B) Description design is underspecified and risks leaking unknown-class semantics.**
Section 3.2 says the description pool is built by “collecting a large number of class names from the internet and generating textual descriptions using templated statements,” and then, for a stage t, the method counts which descriptions are most similar to current data and picks the top  $|C_u^t|$.  This raises two problems:

1. If the global pool already contains *true* or *near-synonym* names of test-time classes, then the model is effectively allowed to *recover class names from unlabeled data* via CLIP, which is a form of label leakage.
2. The paper does **not** show any concrete examples of descriptions, does **not** report pool size, source sites, or filtering rules, and does **not** run the obvious ablation “remove exact/near class names from the pool.”

**(C) Writing / technical precision issues.**
The core distribution-optimization step (Eq. (4)) enforces
$\sum_j z_{i,j} = \frac{1}{N_l + N_u^t}$
and then calls this “probability normalization.” But that is *not* a standard per-sample normalization (one would expect it to be 1); it is then “fixed” later by multiplying back $(N_l + N_u^t)$ before pseudo-labeling. This writing is very confusing. The text around this part also mixes $z_{i,j}$ (line 236), $Z$ (line 256) without re-introducing symbols, and many references are cited in a slightly informal way (e.g., line 28). Moreover, 3.2 says class numbers are estimated (line 215); implementation says they are known (line 374). Overall, the paper feels like a workshop draft.

**(D) Limited novelty relative to CLIP-based GCD / prompt-learning lines.**
The paper itself notes that the proposed optimization is “conceptually similar to prompt learning” and that it simply learns a set of stage-wise semantic vectors and aligns them to CLIP-induced distributions.  This is a reasonable adaptation to the C-GCD protocol, but the ingredients — CLIP feature frozen, text prompts/descriptions, KL alignment to visual prototypes, class-balance constraints, and transductive LP — are all known [1, 2]. What is new is mainly *where* the authors apply it (C-GCD) and that they choose to *reuse* anchors across stages. This is on the thin side unless the empirical section is watertight (which, per (A)–(B), it is not yet).

**(E) Missing ablations for every critical component** (semantic optimization, balanced assignment, LP, expansion).

**(F) No final loss function.**

**(G) No hyperparameter analysis.**

[1] Learning textual prompts for open-world semi-supervised learning.

[2] GET: Unlocking the Multi-modal Potential of CLIP for Generalized Category Discovery

**Questions:**

1. You select top (|C_u^t|) descriptions by counting matches over current-stage unlabeled data. How do you guarantee that the global description pool does **not** contain exact or quasi-exact names of the incoming “unknown” classes? Please provide concrete examples of descriptions and a “no-true-name” ablation.
2. Were the baselines in Table 2 **re-run** with frozen CLIP and the **same** label-propagation inference? If not, can you add such a table, or at least SAC-GCD w/o LP?
3. How sensitive is your method to the size and noise level of the description database (e.g. 1k vs 10k candidates; generic vs fine-grained names)?

---

### Note · Authors · 2025-12-03

I have read and agree with the venue's withdrawal policy on behalf of myself and my co-authors.